# Visualization and Object Detection Based on Event Information

**DOI:** 10.3390/s23041839

**Published:** 2023-02-07

**Authors:** Yinghong Fang, Yongjie Piao, Xiaoguang Xie, Miao Li, Xiaodong Li, Haolin Ji, Wei Xu, Tan Gao

**Affiliations:** 1Changchun Institute of Optics, Fine Mechanics and Physics, Chinese Academy of Sciences, Changchun 130033, China; 2University of Chinese Academy of Sciences, Beijing 100049, China; 3Key Laboratory of Space-Based Dynamic & Rapid Optical Imaging Technology, Chinese Academy of Sciences, Changchun 130033, China

**Keywords:** event information, dynamic vision sensor, visualization, object detection

## Abstract

A dynamic vision sensor is an optical sensor that focuses on dynamic changes and outputs event information containing only position, time, and polarity. It has the advantages of high temporal resolution, high dynamic range, low data volume, and low power consumption. However, a single event can only indicate that the increase or decrease in light exceeds the threshold at a certain pixel position and a certain moment. In order to further study the ability and characteristics of event information to represent targets, this paper proposes an event information visualization method with adaptive temporal resolution. Compared with methods with constant time intervals and a constant number of events, it can better convert event information into pseudo-frame images. Additionally, in order to explore whether the pseudo-frame image can efficiently complete the task of target detection according to its characteristics, this paper designs a target detection network named YOLOE. Compared with other algorithms, it has a more balanced detection effect. By constructing a dataset and conducting experimental verification, the detection accuracy of the image obtained by the event information visualization method with adaptive temporal resolution was 5.11% and 4.74% higher than that obtained using methods with a constant time interval and number of events, respectively. The average detection accuracy of pseudo-frame images in the YOLOE network designed in this paper is 85.11%, and the number of detection frames per second is 109. Therefore, the effectiveness of the proposed visualization method and the good performance of the designed detection network are verified.

## 1. Introduction

A Dynamic Vision Sensor (DVS) is an asynchronous vision sensor inspired by and developed from the abilities of the biological vision system to asynchronously transmit information and to focus on dynamic changes. It differs significantly from the inherent imaging method of traditional optical imaging. Each of its pixels independently detects changes in light intensity and outputs an event when the amount of change exceeds a threshold. The entire pixel array works simultaneously and outputs events asynchronously to form a continuous event stream [1,2,3]. The information contained in the event stream consists of a small amount of data, has a temporal resolution of up to microseconds, and has a high dynamic range. These excellent characteristics of dynamic vision sensors have attracted researchers in different fields to explore and research them and develop their applications, which further promotes their vigorous development.

In many application scenarios, space detection is an application field with high test costs, high difficulty and harsh environments, and the research into applications of dynamic vision sensors in this field is still at a low level [4,5]. With the development of space remote sensing imaging technology in the direction of large field-of-view, high resolution, and ultra-wide format, the data volume of remote sensing images is gradually increasing, which results in higher requirements for data transmission and calculation time, and makes space remote sensing real-time detection of objects particularly difficult. In order to fundamentally reduce the amount of data generated during space detection, it is of great significance to study how to use dynamic vision sensors to complete real-time detection tasks of space remote sensing targets.

This paper aims to study how to reasonably visualize the data acquired by dynamic visual sensors, and design algorithmic models based on the visualized images to achieve target detection. We aim to verify the feasibility of a device based on a dynamic visual sensor completing a target detection task in reality. This is a preliminary study applying the technology to the field of space detection. The following sections will successively introduce the working principle and data structure of the dynamic visual sensor, the research on the visual processing of event information, the research on the target detection algorithm for pseudo-frame images, the production of datasets, and the experimental results and analysis.

## 2. Dynamic Vision Sensor

Unlike the working mode of traditional optical sensors which expose at a fixed frame rate and obtain the exposure of all pixels at the same time, each pixel of a DVS detects the light intensity in the environment independently and then records and compares it in the form of logarithm of light intensity. When the detected change in light intensity reaches a certain threshold, a data packet called an event containing the time, the address of the pixel, and the lightness increase (ON) or decrease (OFF) is outputted. All pixels on the DVS asynchronously detect light intensity at the same time, and continuously output such events to generate an event stream. A single pixel circuit structure diagram of a DVS is shown in Figure 1. The circuit consists of a logarithmic photoreceptor, a differential circuit, and two comparators [6,7,8].

Typically, an event can be described as:(1)$$e_{k}=\left(x_{k},t_{k},p_{k}\right)$$
where $x_{k}={\left(x_{k},y_{k}\right)}^{T}$ expresses the address where it occurred, $t_{k}$ expresses the time when it occurred, and $p_{k}\in \left\{+1,-1\right\}$ expresses its polarity. $p_{k}=+1$ represents a brightening event, and $p_{k}=-1$ represents darkening event. The logarithm of light intensity at $x_{k}$ is expressed as:(2)$$L(x_{k},t_{k})=\log(I(x_{k},t_{k}))$$
where I is the light intensity.

If an event is fired, this means that the difference of the logarithm of the light intensity at the corresponding pixel between this event and the last event exceeds the preset threshold ±*C*, which is expressed as:(3)$$\Delta L(x_{k},t_{k})=p_{k}C$$
and there is:(4)$$\Delta L(x_{k},t_{k})=L(x_{k},t_{k})-L(x_{k},t_{k}-\Delta t_{k})$$
where $\Delta t_{k}$ is the time since the last event fired at $x_{k}$.

The mathematical model of the above-mentioned generation events is established under the ideal situation without noise, but actually the sensor is affected by the noise caused by the external and internal circuits of the photoreceptor. By setting the threshold *C*, the sensitivity of the DVS to noise can be adjusted: the larger the threshold *C*, the less sensitive the DVS is to noise, but the events captured by the DVS will decrease accordingly; the smaller the threshold *C*, the more events captured by the DVS, but the signal-to-noise ratio will be seriously reduced.

The changes in lightness detected by the sensor can be caused by changes in lightness of the scene, by movement of objects in the scene, or by the sensor itself. When the lightness of the scene changes faster, the objects in the scene or the sensor itself move faster, the more events are generated by the DVS per unit time. Depending on differences in hardware and design, the DVS’s output rate can be as high as 1200 MEPS (events per second, EPS), the delay can be as low as 1μs, and the dynamic range can reach 140 dB. Such characteristics provide DVSs great advantages in applications such as capturing high-speed moving objects, responding to high dynamic range lighting environments, and realizing low-latency control.

With the development of hardware technology and the catalysis of use requirements, DVSs have gradually evolved into two further types: asynchronous time-based image sensors (ATIS) [9], and dynamic and active pixel vision sensors (DAVIS) [10,11]. They have different characteristics and different usage scenarios, and also have their own drawbacks. Table 1 is a comparison of the characteristics and applications of the three sensors.

## 3. Visualization of Event Information

Considering the data structure of a DVS, compared with image information, event information has some specific characteristics. It contains less information, is relatively independent, and has a low degree of visualization, large noise influence, high temporal resolution, etc. In order to further study the capabilities and characteristics of event information to represent the target in view of these characteristics, we propose a visual processing algorithm of event information with adaptive temporal resolution.

### 3.1. Pseudo-Framing of Event Information

A DVS detects changes of light intensity and outputs event information in the form of asynchronous event streams. However, a single event contains little information and has poor visibility, which makes subsequent processing of event information difficult. In order to visualize event information, according to the correlation between events, events within a period of time are projected on time slices to make frame images, which are called pseudo-frame images, and this process is called pseudo-framing [12]. In the process of pseudo-framing, information such as object movement features and shape features carried by event information can be displayed in pseudo-frame images, which is more convenient for subsequent feature extraction and target detection. Figure 2 shows the process of pseudo-framing the event stream.

According to the positive and negative polarity, the event information can be divided into two parts: brightening and darkening. If the motion factor is from the photographed object, the brightening event will generate in front of the object, and the darkening event will generate behind the object; if the motion factor is the DVS, the brightening and darkening events will generate in the opposite position. Therefore, the following observations can be made: firstly, there is a temporal correlation between the brightening and darkening events; second, the brightening and darkening events can roughly describe the outline characteristics of the object; and third, it is necessary to pay attention to the difference between the brightening and darkening events generated when the DVS and the object are in motion.

Based on the above analysis, according to the event polarity, the event stream within a period of time is projected on two all-zero matrices with the same spatial resolution as the DVS. In these two matrices, a value of 1 is assigned to the position where there is an event. In this way, the changed lightness information and position information carried by the event can be displayed in the form of a binary image. Then, two decomposed images that can express the brightening and darkening events separately are obtained. It should be noted that there may be a few events at a certain coordinate position in the matrix during this period. In this case, the previous events should be overwritten and only leave the last event.

The two event decomposed images in the same time interval jointly describe the state and characteristics of the object during this period. For a clearer display, the two decomposed images with two channels are converted into an initial pseudo-frame image which is a three-channel color image. In the initial pseudo-frame image, the brightening area is displayed in red, the darkening area is displayed in green, and if there are both brightening and darkening events at a position, it is displayed in yellow. In this way, the process of pseudo-framing to the event information is completed. Figure 3 is the result of pseudo-framing to the event stream.

### 3.2. Denoising and Filtering of Pseudo-Frame Images

Each pixel of a DVS independently responds to changes in ambient lightness to continuously output information at a level of microseconds. In this process, the DVS tends to generate false events in the circuit when it is affected by the temperature, which is also called generated noise [13]. In addition, the non-uniform changes of the target lightness also lead to many defects in the target area. A large amount of noise and many defects not only affects the shape characteristics of the target, which has a great impact on the adaptive temporal resolution processing of event streams and the effectiveness of target detection, but also consumes a lot of computing resources and results in increased data processing time.

Due to the spatial continuity of the object, the lightness changes produced during its movement must also be continuous. In other words, there is a temporal and spatial correlation between real events, whereas the generation of false events is generally random. In the pseudo-frame image, the noise is reflected as scattered points, and the defects are represented as holes and slits in the target area, as shown in Figure 4. According to this feature, the number of events in the vicinity of the target pixel can be used to determine whether there is a real event at the pixel position, and then achieve the purpose of removing noise, repairing holes and slits.

Median filtering is a process to statistically sort the target pixel value and its adjacent pixel values, and takes the median value as the pixel value of the target pixel position. This processing method is consistent with the above analysis, and is a very suitable existing filtering method. Figure 5 is the result obtained after processing the pseudo-frame image using a median filter with a size of 3 × 3.

It can be observed from Figure 5 that, for large targets which have clear contour features, median filtering has a good effect on denoising and repairing, but it also filters out too much real information of small targets. Therefore, for small targets, this paper uses the method of nearest neighbor filtering. Nearest neighbor filtering counts the number N of real events at the adjacent pixel positions of the target pixel and determines the value of the target pixel position by judging the size of N. Figure 6 is a flowchart of nearest neighbor filtering. 

Figure 7 shows the results obtained by filtering pseudo-frame images using the nearest neighbor filter with parameters [A, B, C] = [5 × 5, 3, 14]. It can be seen that, compared with the median filter, the nearest neighbor filter can retain more real information of small targets.

### 3.3. Adaptive Temporal Resolution

Pseudo-framing of event streams is an effective method to exploit event information in practice. Currently, there are mainly two pseudo-framing schemes [14,15]: the first is constant time interval (CTI), which is to project all events in the time interval onto time slices. A series of pseudo-frame images obtained by this method have a constant frame rate that is similar to traditional frame images, as described in Equation (5), where *I* is the matrix of the pseudo-frame image, *A* is the all-zero matrix, *N* is the number of events, and *T* is the time interval.
(5)$$\begin{array}{l} I=\sum_{i=1}^{N}\left[(|p_{i}|)|A(x_{i},y_{i})\right],\\ N:\;\;t\leq t_{N}<t+T \end{array}$$

The second pseudo-framing scheme is the constant number of events (CNE), which projects the same number of events onto the time slice. A series of pseudo-frame images obtained by this method have different frame rates, as described in Equation (6).
(6)$$I=\sum_{i=1}^{N}\left[(|p_{i}|)|A(x_{i},y_{i})\right]$$

Figure 8 shows the pseudo-frame images obtained by these two methods under different motion states. For the method with a constant number of events, when the size of the target or the complexity of the background changes, the clarity of the target in the pseudo-frame image cannot be guaranteed, such as the first and fourth images in (b). For the method of constant time intervals, when the relative speed of the target and the device changes greatly, the distribution of events on the time slice is redundant and sparse in the high-speed and low-speed states, respectively, which leads to the inability to effectively distinguish the target in the image, such as in the first and fourth images in (c).

The states of the environment and the target are complex and changeable in actual capturing, but there are some defects in pseudo-framing methods using both the fixed time intervals and the fixed number of events, which make it difficult to detect and monitor the state of objects in real time. On one hand, for fast-moving targets, we hope to take advantage of the high temporal resolution of the DVS for clear imaging; on the other hand, for slow-moving targets, we hope to reduce the number of images as much as possible on the basis of clear imaging so as to reduce the amount of data. It can be seen that a better imaging method is to use the characteristics of the event stream to timely adjust the temporal resolution of pseudo-frame images according to the motion state of the targets. Based on the above analysis, this paper proposes a method that can adaptively update the time resolution of the pseudo-frame image according to the actual motion speed of the captured object, which can realize the adaptive time resolution in the processing of event information visualization.

According to the description in Section 3.1, there are only red areas (only brightening events), green areas (only darkening events), yellow areas (both brightening and darkening events) where there are events in the initial pseudo-frame image. Considering that the front end of the moving target generates a brightening event, and its rear end generates a darkening event, assuming there is no noise, the meaning of the yellow area is that both the front end and the rear end of the moving target have passed the same pixel position in a certain time. If there are many yellow pixels (also called overlapping points) in the pseudo-frame image during this period, it means that the captured object has a relatively high speed during this period when the lighting conditions remain unchanged. Therefore, it can be concluded that the number of overlapping points in the initial pseudo-frame image can indirectly reflect the speed of the object. This is also an important reason to separate brightening and darkening events when conducting the pseudo-frame processing of event information.

According to the above analysis, it can be concluded that given an initial time interval, if the object moves too fast, a large number of overlapping points are generated during this period, which makes it difficult to distinguish the characteristics of the object in the initial image. Conversely, if the object moves too slowly, too few events are generated during this period, with the result that these events are unable to accurately represent the characteristics of the object in the initial pseudo-frame image. Therefore, the number of overlapping points can be used to define the temporal resolution of the initial pseudo-frame image. The implementation of the adaptive temporal resolution visualization method is described in Equation (7), where $T_{0}$ is the minimum time unit, $F_{a}$ and $F_{b}$ are the upper and lower limits of the decision range, and $N_{e}$ is the number of overlapping points.
(7)$$\begin{array}{l} I=\sum_{i=1}^{N}\left[(|p_{i}|)|A(x_{i},y_{i})\right],\\ N:\;\;t\leq t_{N}<t+T+\Delta t,\\ \Delta t=\left\{\begin{array}{c} -T_{0}\\ 0\\ T_{0} \end{array}\right.\begin{array}{c} ,N_{e}>F_{b}\\ ,F_{a}\leq N_{e}\leq F_{b}\\ N_{e}<F_{a} \end{array} \end{array}$$

The specific implementation method is introduced below:

The judgment range $\left(F_{a},F_{b}\right)$ is set and a minimum time unit $T_{0}$ microseconds is set according to the shooting environment.

Considering the relationship between *N* (the number of overlapping points in the initial pseudo-frame image) and the judgment range, if $N_{e}\geq F_{b}$, the brightening and darkening events overlap substantially, which means the target is moving relatively fast, and the events are crowded. In this case, the initial pseudo-frame image is discarded and the time interval is reduced by a specified minimum time unit, that is, the time interval is updated to $T_{g}-T_{0}$, and pseudo-frame processing is continued on this part and subsequent part events. If $N_{e}\leq F_{a}$, it means that there is no overlapping phenomenon in the initial image, the target object is moving relatively slowly, and the number of brightening and darkening events is small. In this case, the initial pseudo-frame image is discarded and the time interval is increased by a specified minimum time unit, that is, the time interval is updated to $T_{g}+T_{0}$, and pseudo-frame processing is continued on this part and subsequent part events. If $F_{a}\leq N_{e}\leq F_{b}$, it means that the proportion of events is reasonable. This initial pseudo-frame image is retained and the generated image is obtained.

Through the above processing, a series of frame images with relatively reasonable time resolution can be obtained from the event stream, which not only ensures the clear presentation of the status and characteristics of the captured objects in the pseudo-frame image, but also reduces the relative amount of data and calculation. Figure 9 shows pseudo-frame images obtained with the method of adaptive temporal resolution (ATR). It can be seen that the clarity of the target features is improved when the target size and speed change significantly.

### 3.4. Section Summary

This section proposes an event information visualization method that can achieve adaptive temporal resolution, which can ensure timely updates to the temporal resolution of a series of pseudo-frame images according to changes in the target motion state. It not only takes full advantage of the high temporal resolution of dynamic vision sensors, but also reduces the amount of data. The storage size of the pseudo-frame image obtained by the above method is generally less than 20 KB, and generally only contains the outline information of the object. It can be seen that the pseudo-frame image is an image with simple features consisting of a small amount of data. Figure 10 represents the flowchart of the event information visualization method proposed in this paper.

## 4. Object Detection for Pseudo-Frame Images

The purpose of obtaining the pseudo-frame image is to express the target information carried by the event data in a more intuitive way. However, the effectiveness of such images in practical applications still needs to be verified. Therefore, depending on the background of the subject, we investigate fast target detection using pseudo-frame images.

The target detection algorithm of traditional images is quite mature, and the application of deep learning even results in the target detection accuracy of machines surpassing that of humans. Pseudo-frame images are similar to traditional images in structure and have the basic characteristics of traditional images, but compared with traditional images, they have small data volumes, simple features, and generally only contain target contour information. In this section, we design and optimize a target detection algorithm based on these characteristics.

Compared with the traditional target detection algorithm which sequentially uses algorithms such as HOG, LBP, and Harris to extract features from images, and classifiers such as SVM and Decision Tree to learn and classify features, the target detection algorithm based on the convolutional neural network has better efficiency and accuracy, and the YOLO algorithm is one of the best developed ones. In an experiment conducted with quickly moving playing cards, Qiu Zhongyu [15] proved that for the detection effect of pseudo-frame images, the target detection method based on convolutional neural network feature extraction is better than the manual feature extraction target detection method. However, the detection accuracy obtained by the Event-YOLO detection model he proposed was only 78.3%. Since the YOLO algorithm was proposed by Joseph Redmon et al. in 2016, researchers have improved and optimized the model according to different usage scenarios to meet their actual needs, which has enriched the network structure of YOLO and produced many variations of the model. The YOLO network has a simple structure, a small amount of calculation, and has good migration and generalization performance. Therefore, we use the YOLO network as a prototype to improve the model, and verify its detection accuracy and the real-time performance of the target in the pseudo-frame image. The main idea of the network design is, based on the YOLO network, to design and optimize the structure of each part of the network. To ensure sufficient detection accuracy, the model size is reduced as much as possible and the detection speed is increased. The optimization method of each part of the network structure is as follows [16]:

For the backbone network, we use the residual network to complete the feature extraction. Pseudo-frame images contain less information and features of the target. Although more convolution operations can enrich the semantic information of the image, it is also easy to cause the original information of the target to be lost. The unique structure of the residual network can reduce the loss of information when extracting features. Therefore, based on the Darknet53 network and the residual network, this paper designs the backbone network structure shown in Figure 11.

For the neck network, we use the optimized FPN (Feature Pyramid Network) module to complete the feature fusion. FPN can realize multi-scale information fusion in the network which integrates low-level detailed information and high-level semantic information, improves the robustness of the model for spatial layout and object variability, and only results in a small increase in calculation. Therefore, using the FPN module can greatly improve the model performance. At the same time, convolutional layers are added to the FPN module to enhance the effect of feature fusion. Figure 12 shows the feature fusion structure of optimized FPN.

For the head network, we use convolutional layers for feature decoding and prediction. The fully connected layer has poor nonlinearity, a large amount of calculation, and it also limits the size of the image. Therefore, convolutional layers with good nonlinearity and requiring less calculation are used to complete the feature decoding. In addition, a 1 × 1 convolution is used to complete target classification and position prediction, where the branch of predicting categories uses the Softmax Activation Function, and the branch of predicting bounding boxes uses the Sigmoid Activation Function. Figure 13 shows the head network structure.

For the loss function part of the model, the predicted value of Class and Objectness is calculated using the cross-entropy (CE) function shown in Equation (8) to calculate the loss. The predicted value of Bbox is calculated using the mean square error (MSE) function (Equation (9)). Equation (10) is the calculation method of the total loss.
(8)$$CE=-\sum_{i=1}^{n}\left(y_{i}\log({\hat{y}}_{i})+(1-y_{i})\log(1-{\hat{y}}_{i})\right)$$
(9)$$MSE=\frac{1}{n}\sum_{i=1}^{n}{\left(y_{i}-{\bar{y}}_{i}\right)}^{2}$$
(10)$$L=L_{cls}+L_{obj}+0.1\times (L_{t_{x},t_{y}}+L_{t_{w},t_{h}})$$

Thus, the overall optimization design of the detection network is complete. Figure 14 shows the structure of the YOLOE network model designed in this paper.

## 5. Experiment and Analysis

### 5.1. Experiment Environment

The experimental equipment used in this research is DAVIS 346, a product of the Inivation company. Its main parameters are as follows: the spatial resolution is 260 × 346, the time resolution is 1 microsecond, the dynamic range is about 120 dB, and the event output rate is 12 MEPS. Figure 15 shows the actual DAVIS 346. Figure 16 is an experiment image when using DAVIS346 for data collection. In addition, the experimental conclusions recorded in this paper are all obtained on a computer with a i9-11900H CPU and a RTX3070 GPU.

### 5.2. Creating Datasets

The current public datasets of event information cannot be used to verify the effect of the pseudo-framing method and the performance of the detection network proposed in this paper. Therefore, we designed experimental plans, collected event information, and created two datasets for the research in this paper.

Dataset 1: In order to fully verify the feature preservation effect of the pseudo-framing method for event information proposed in this paper, scissors were used as the experimental object to collect event information for different shapes, sizes, and speeds. The three pseudo-framing methods introduced in Section 3 were used to obtain pseudo-frame images from the event information, and 400 images with clear features were then selected for labeling to construct a training set. The scissors were moved in an irregular manner and a section of event information was collected. The three pseudo-framing methods were used to obtain the same number of pseudo-frame images. The resulting images were then labelled to form a test set. The YOLOv3 network was used for training, testing, and comparing the obtained test results.

Dataset 2: In order to verify the detection performance of the improved network in this paper, pedestrians and vehicles were used as experimental objects to collect data in two ways: static shooting and moving shooting. Static shooting is shooting with a fixed camera, and the information obtained only includes moving people and cars. The background is simple. Moving shooting is shooting with a moving camera, and the information obtained includes all objects in the field of view. The background is complex. The pseudo-framing method was used with adaptive temporal resolution to obtain pseudo-frame images from the collected event information. Then 1000 images were selected for labeling to form a dataset, in which the ratio of the training set, test set and verification set is 8:1:1. Due to the different datasets used in experiments, the experimental results in similar papers cannot be directly quoted for comparison. In this part of the experiment, several representative public detection algorithms are selected as experimental comparison items, including the improved YOLOv1 model, YOLOv2 model and YOLOv3 model. These were trained and tested on the datasets together with the YOLOE model, and their detection accuracy, detection rate and model size are compared.

### 5.3. Experimental Results and Analysis

(1) Using dataset 1, we tested the AP values of pseudo-frame images obtained by three different visualization processing methods under the obtained training model, and the results are shown in Table 2. According to these results, the AP of the image obtained by the ATR was 5.11% and 4.74% higher than that obtained with CTI and CNE, respectively. It can be concluded that the feature preservation effect of the event information visualization method with adaptive temporal resolution proposed in this paper is superior to the other two methods.

(2) Using dataset 2, we verified the detection performance of the YOLOE detection network. Table 3 records the test values of the mean detection accuracy (mAP), detection frame rate (FPS) and model size with the different networks. According to these results:(a)The model size is directly proportional to the mAP and inversely proportional to the FPS;(b)The improved YOLOv1 and YOLOv2 have lower mAP than YOLOv3 and YOLOE, but have higher FPS;(c)YOLOv3 has the highest mAP, the largest model size, and the lowest FPS;(d)YOLOE’s mAP is 85.11%, 6.84% less than YOLOv3; YOLOE’s FPS is 109, 32 more than YOLOv3; YOLOE’s model size is 163 MB, 72 MB less than YOLOv3.

The improved YOLOv1 and YOLOv2 are suitable for applications that demand higher detection speed and do not require high detection accuracy. YOLOv3 is suitable for applications that demand higher detection accuracy and do not require detection speed. YOLOE has a balanced performance in detection accuracy and detection speed, and is suitable for applications that have certain requirements for detection accuracy and detection speed. Therefore, it can be concluded that the YOLOE target detection network designed in this paper can maintain a high detection speed when it has high detection accuracy, so it has better comprehensive detection performance. At the same time, it can be concluded that pseudo-frame images can provide good detection results, even under a small-scale deep learning model.

## 6. Discussion and Conclusions

This paper aims to study the processing methods of event information and the application effects in the field of target detection. Firstly, this paper introduces the working principle of the dynamic vision sensor and analyzes the structural characteristics of the event information. Compared with image information, event information is characterized by less information, lower visualization, greater influence of noise, and higher temporal resolution. Secondly, in order to further study the capability of event information to characterize targets, we propose an event information visualization method with adaptive temporal resolution. The algorithm sequentially performs pseudo-framing, denoising and filtering, and adaptive temporal resolution on the event information to obtain a series of pseudo-frame images that can represent the characteristics of the target, making the event information more intuitive. Thirdly, in order to explore whether the pseudo-frame image can efficiently complete the task of target detection given that pseudo-frame images include small data volumes, simple features, and generally only contain contour information of the target, we designed the YOLOE detection network to verify the detection effect on pseudo-frame images. Finally, we designed an experimental scheme, collected data, and constructed two datasets for experimental verification.

Through the experimental testing, we determined that the detection accuracy of the image obtained by the event information visualization method with adaptive temporal resolution was 5.11% and 4.74% higher than that obtained using methods with constant time intervals and a constant number of events, respectively. The average detection accuracy of pseudo-frame images in the YOLOE network designed in this paper was 85.11%, and the number of detection frames per second was 109. It was verified that the event information visualization method proposed in this paper is effective, and that the designed detection network has good detection ability. At the same time, it was demonstrated that despite the small amount of data and simple features, pseudo-frame images can obtain good detection results, even under a small-scale deep learning model.

Although the event information visualization method with adaptive temporal resolution makes good use of the high temporal resolution of event data, this processing method still does not fully utilize the characteristics of a DVS. The data processing method directly taking events as the object retains the characteristics of a DVS to the greatest extent, and it will have a broader development space in the future. In addition, when detection backgrounds and detection targets change, the detection performance of the target detection model introduced in this paper will fluctuate greatly because it is designed based on the research background of this paper. It may not even be applicable in some scenarios, such as those with large and complex backgrounds, scenarios focusing on fine textures of the target surface, or scenarios with small and jumbled targets. Therefore, different models need to be established for different detection tasks to achieve the best results.

## 7. Patents

A patent application is pending with the Changchun Institute of Optics, Fine Mechanics and Physics, Chinese Academy of Sciences.

## Figures and Tables

**Figure 1 sensors-23-01839-f001:**
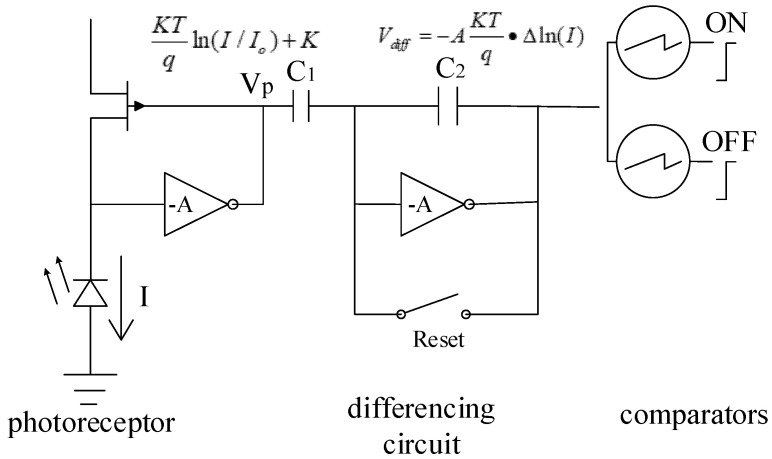
Circuit structure of a DVS pixel [7].

**Figure 2 sensors-23-01839-f002:**
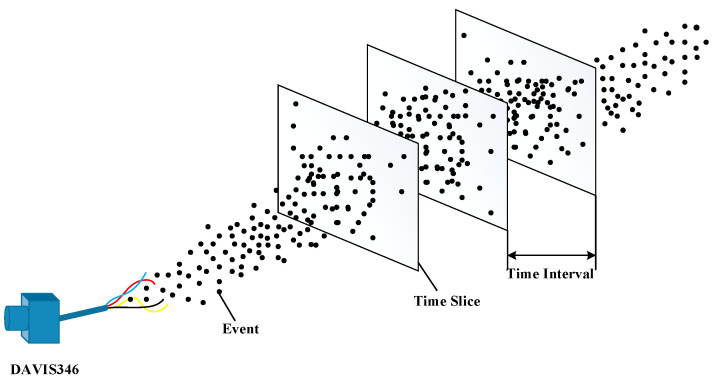
Pseudo-framing process of event stream.

**Figure 3 sensors-23-01839-f003:**
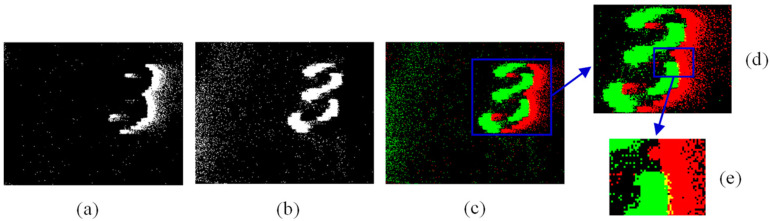
Moving the device quickly to the right to capture a stationary target and performing pseudo-framing on the resulting event stream: (**a**) is the brightening decomposed image, (**b**) is the darkening decomposed image, (**c**) is the initial pseudo-frame image, and (**d**,**e**) are enlarged detail images.

**Figure 4 sensors-23-01839-f004:**
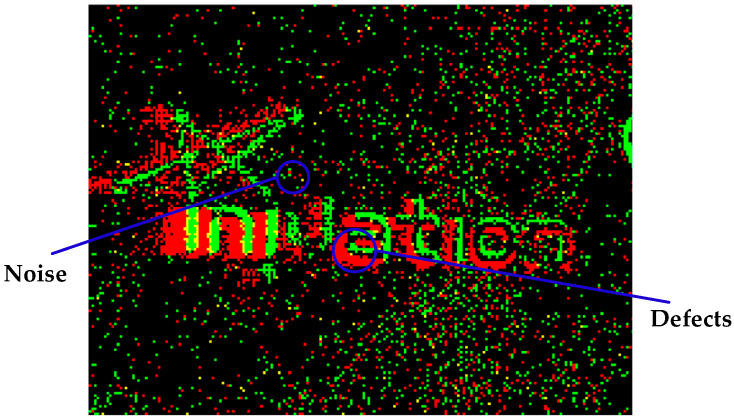
Noise and defects.

**Figure 5 sensors-23-01839-f005:**
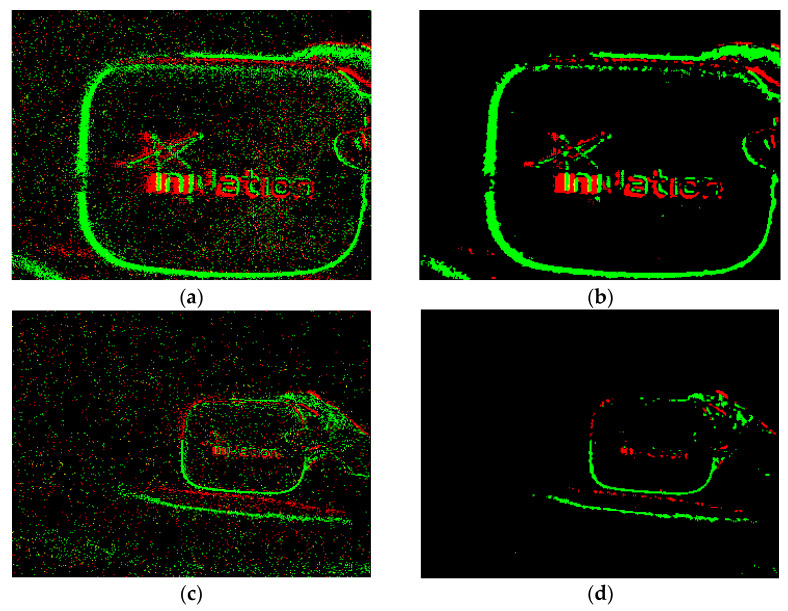
Comparison of the effects of median filtering. (**a**,**b**) show the filtering effect on a large target, (**c**,**d**) show the filtering effect on a small target. (**a**,**c**) are the original images, and (**b**,**d**) are the filtered images.

**Figure 6 sensors-23-01839-f006:**
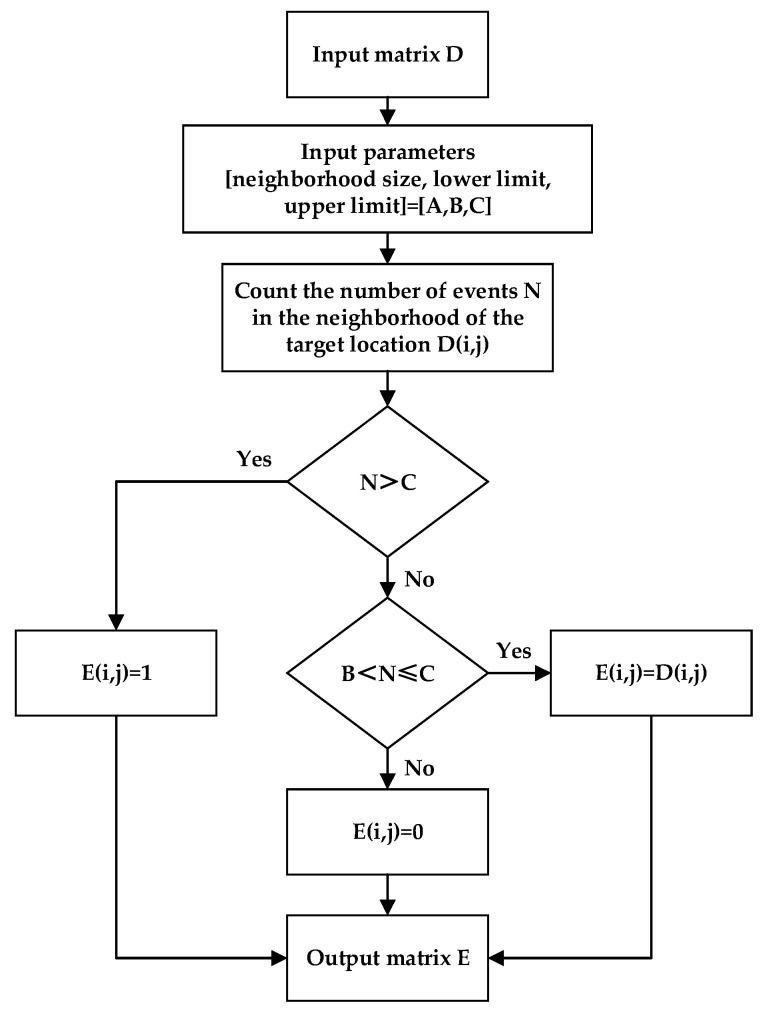
Flowchart of nearest neighbor filtering.

**Figure 7 sensors-23-01839-f007:**
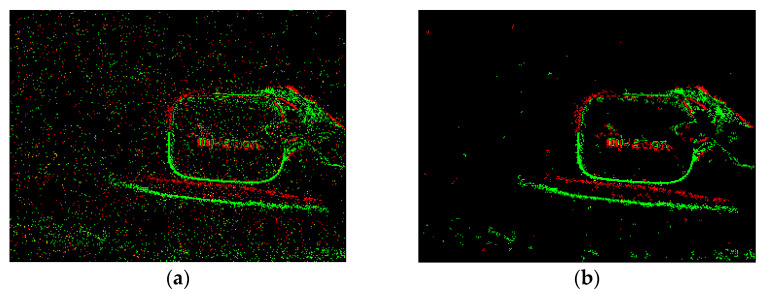
The filtering effect of the nearest neighbor filter on a small target. (**a**) is the original image, and (**b**) is the filtered image.

**Figure 8 sensors-23-01839-f008:**
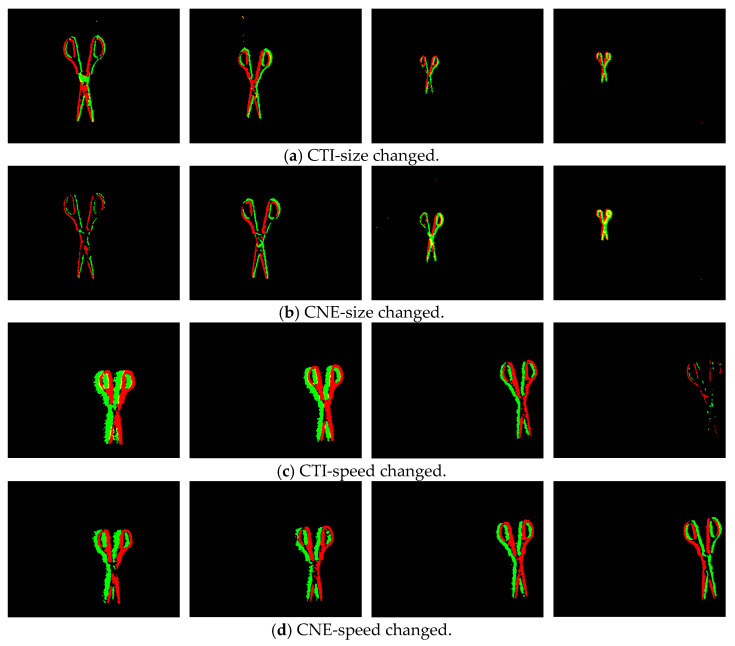
Pseudo-frame images obtained by the two methods in different motion states.

**Figure 9 sensors-23-01839-f009:**
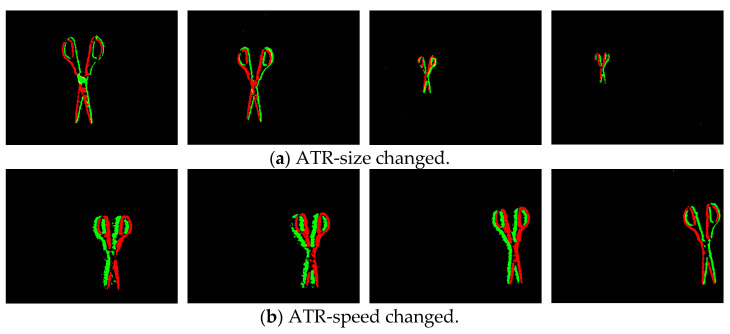
Pseudo-frame images obtained by the method of adaptive temporal resolution in different motion states.

**Figure 10 sensors-23-01839-f010:**
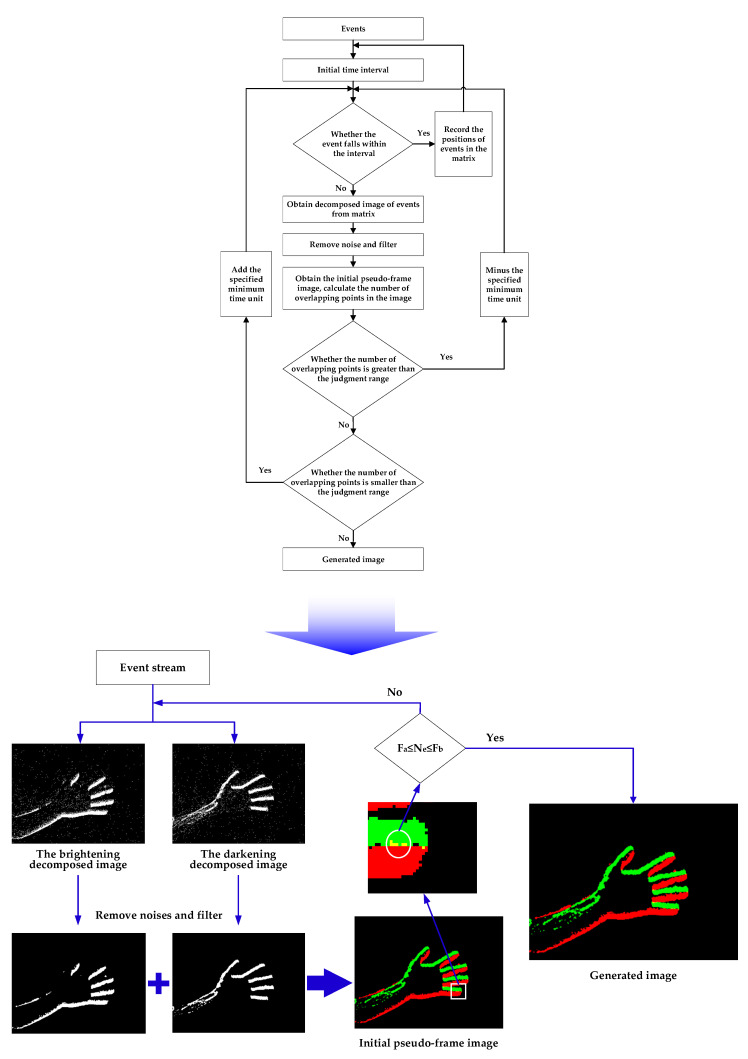
Algorithm flowchart of event information visualization method with adaptive time resolution.

**Figure 11 sensors-23-01839-f011:**
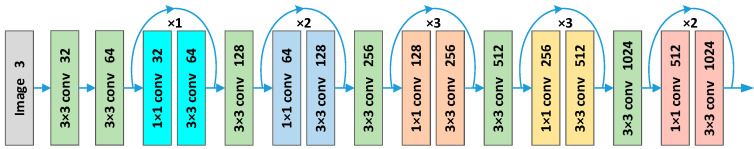
Structure of backbone network.

**Figure 12 sensors-23-01839-f012:**
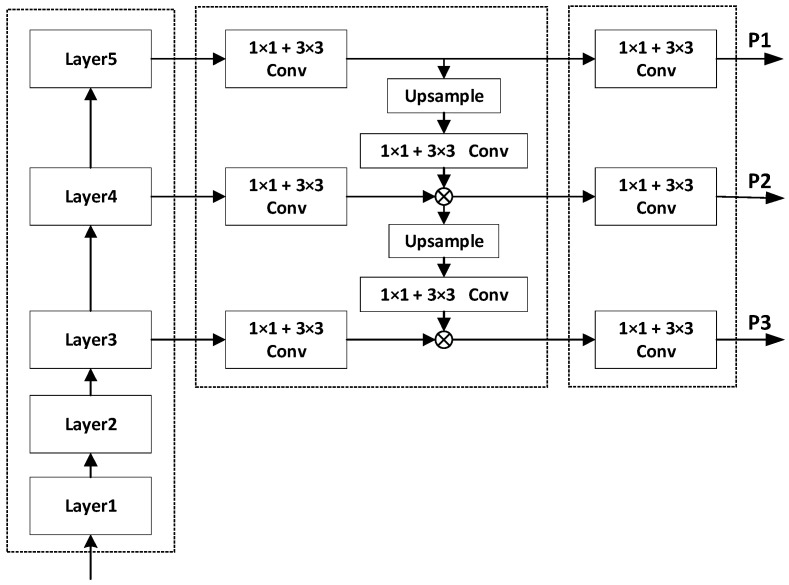
Feature fusion structure of the optimized FPN.

**Figure 13 sensors-23-01839-f013:**
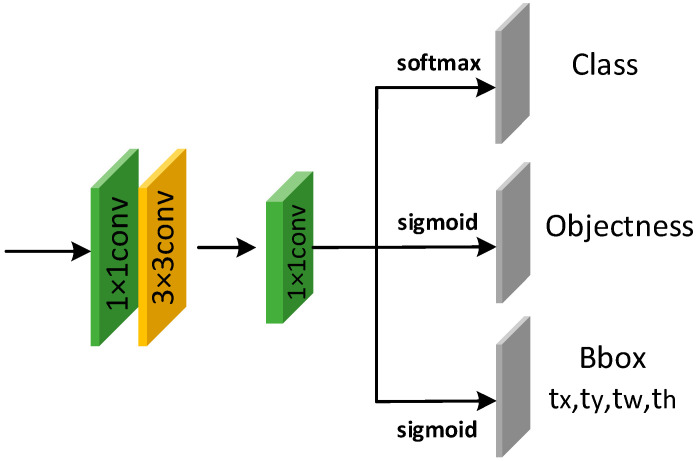
Structure of head network.

**Figure 14 sensors-23-01839-f014:**
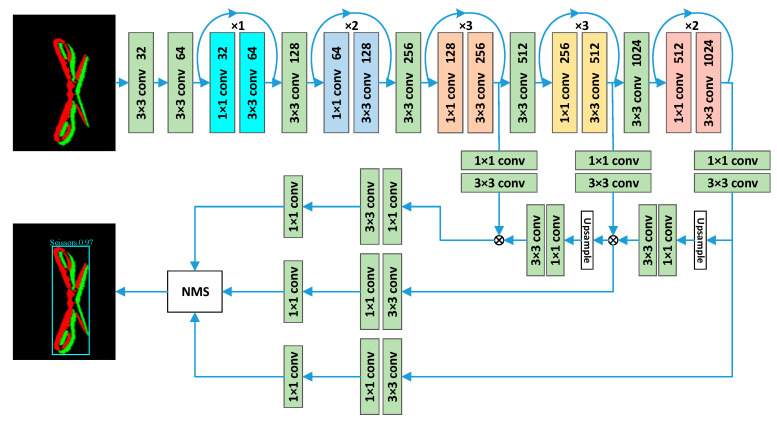
Structure of YOLOE network model.

**Figure 15 sensors-23-01839-f015:**
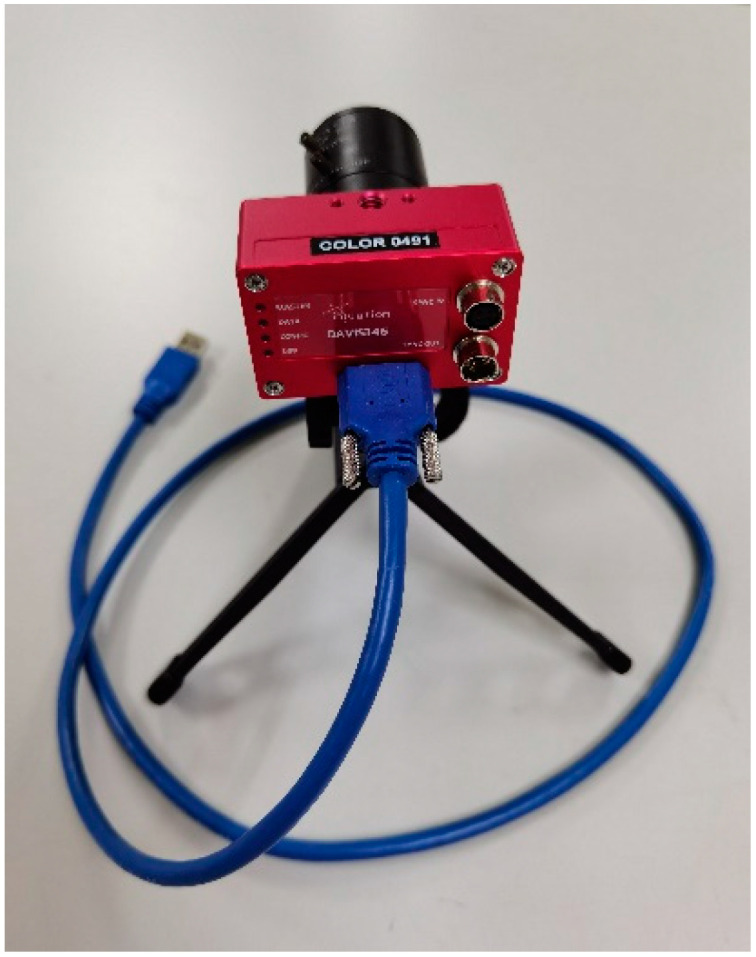
DAVIS 346.

**Figure 16 sensors-23-01839-f016:**
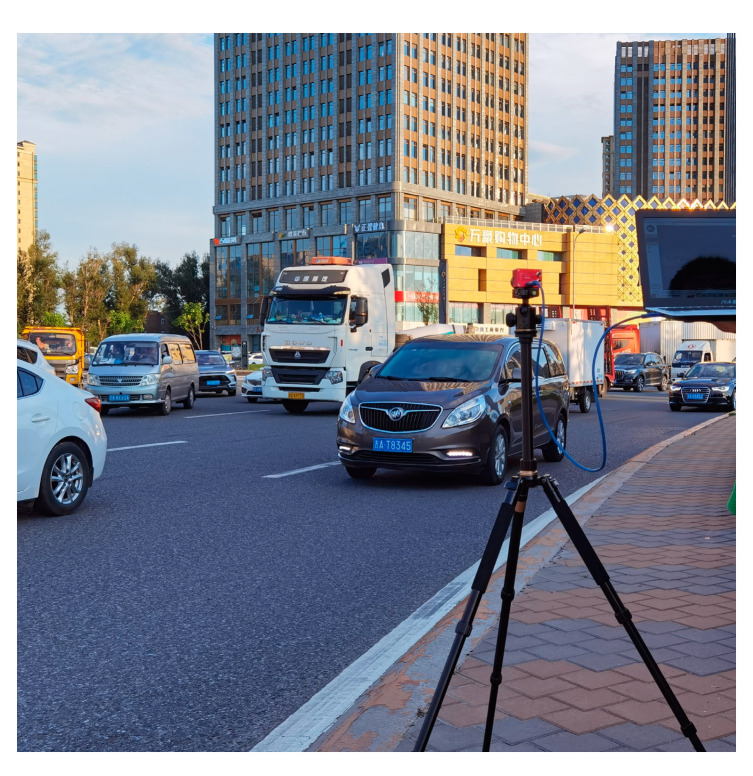
Collection process of event data.

**Table 1 sensors-23-01839-t001:** Comparison of characteristics and applications of three sensors.

Class	Characteristics	Applications
DVS	Only outputs event information and can retain the advantages of low latency, low power consumption, and high dynamic range, but cannot output any grayscale information, and has less visibility.	Generally used in situations with low requirements for visualization and high requirements for avoiding delay, such as tracking of high-speed moving objects, particle counting, motion monitoring, real-time positioning, etc.
ATIS	Can output event information and grayscale information of pixels where the light intensity has changed, and can retain the advantages of low latency, low power consumption, and high dynamic range, and therefore, has partial visibility. However, when the ambient lightness changes slowly, it is prone to problems such as abnormal exposure, information loss, etc.	Generally used in situations where the ambient lightness changes frequently and there is high-speed movement, such as real-time monitoring in industrial manufacturing.
DAVIS	Can output event information and grayscale information, which can be combined, so it has high temporal resolution and good visibility, and can obtain detailed information from moving targets. However, it suffers from APS camera flaws, such as redundant information, low time resolution, and low dynamic range.	It is generally used in situations with high requirements for visualization and a small dynamic range, such as target recognition, target detection, target tracking and positioning, especially in the field of robots and unmanned driving.

**Table 2 sensors-23-01839-t002:** Test results of different visualization methods.

	CTI	CNE	ATR
AP	83.76%	84.13%	88.87%

**Table 3 sensors-23-01839-t003:** Test results of different network models.

	mAP	FPS	Model Size (MB)
YOLOv1-Resnet18	56.36%	272	57
YOLOv1-Resnet34	62.84%	200	95
YOLOv1-Resnet50	69.11%	104	165
YOLOv2	77.86%	160	137
YOLOv3	91.95%	77	235
YOLOE (ours)	85.11%	109	163

## Data Availability

The data presented in this study is available on request to the correspondent author with appropriate justification.

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
