# Peer review of "Visualization and Object Detection Based on Event Information"

_sensors, 2023, doi:10.3390/s23041839_

Round 1
Reviewer 1 Report
The paper has several situations of typos that must be reviewed for a better understanding of the proposal. Some figures appear with low resolution and very small type. This reviewer considers that one or several fully structured demonstrative experiments, where the results to be reached are known, are necessary. These same experiments must be compared with other proposals. And additionally, of course, it must be included the results presented in Section 5.
Author Response
Point1: The paper has several situations of typos that must be reviewed for a better understanding of the proposal. Some figures appear with low resolution and very small type. This reviewer considers that one or several fully structured demonstrative experiments, where the results to be reached are known, are necessary. These same experiments must be compared with other proposals. And additionally, of course, it must be included the results presented in Section 5.
Responses1:
Thank you so much for taking the time to read the article and suggest revisions. I've changed some typos that were found, and I've changed Figure 10 to an image with a larger font.
I agree with your opinion strongly that formulating a structured experimental plan can make the content of the paper more convincing. There are also some comparative experiments in the paper: for the method of event information visualization introduced in the paper, I compared it qualitatively and quantitatively with the method of constant time interval and the method of constant number of events; since the dataset used in the experiment was made according to the visualization method proposed in this paper, the experimental data in similar papers cannot be directly used for comparison. So, for the target detection model based on the convolutional neural network designed in this paper, I chose several publicly available representative convolutional neural network models as the comparison items of the experiment.
Based on your suggestion, I have realized that the explanation and description of the experimental results in the paper are not clear enough, and the introduction of other similar studies is lacking, so I have made some modifications in the paper.
Some modifications are listed below:
1、In section 4, I have added an introduction to a similar study. “By quickly moving the playing cards to conduct experiments, Qiu Zhongyu [15] proved that for the detection effect of pseudo-frame images, the target detection method based on convolutional neural network feature extraction is better than the manual feature extraction target detection method. However, the detection accuracy obtained by the Event-YOLO detection model he proposed is only 78.3%.”
2、In section 5.2, I have added some explanations about choosing the experimental comparison item. “Due to the different datasets used in the experiments, the experimental results in similar papers cannot be directly quoted for comparison. In this part of the experiment, several representative public detection algorithms will be selected as experimental comparison items, including the improved YOLOv1 model, YOLOv2 model and YOLOv3 model. Respectively train and test on these models and the YOLOE model,”
3、In section 5.3, I have added some descriptions about the comparison of the experimental results with other methods:
(1)Add “the AP of the image obtained by the ATR was 5.11% and 4.74% higher than that the method with CTI and CNE obtained, respectively.” to the first experimental analysis.
(2)Add “
- The model size is directly proportional to the mAP and inversely proportional to the FPS.
- The improved YOLOv1 and YOLOv2 have lower mAP than YOLOv3 and YOLOE,but have higher FPS.
- YOLOv3 has the highest mAP, the largest Model size, and the lowest FPS.
- YOLOE's mAP is 85.11%, 6.84% less than YOLOv3; YOLOE's FPS is 109, 32 more than YOLOv3; YOLOE's Model size is 163, 72 less than YOLOv3.
The improved YOLOv1 and YOLOv2 are suitable for applications that pay more attention to the detection speed and do not require high detection accuracy. YOLOv3 is suitable for applications that pay more attention to detection accuracy and do not require detection speed. YOLOE has a balanced performance in detection accuracy and detection speed, and is suitable for applications that have certain requirements for detection accuracy and detection speed. Therefore,” to the second experimental analysis.
Has been modified in the manuscript, please check.

Reviewer 2 Report
The paper is well written and the results are presented in the right form. It is pretty easy to follow the article. The presented approach could be interested for the readers, however the results obtained cannot be considered as a breaking-through technology.
Author Response
Point1: The paper is well written and the results are presented in the right form. It is pretty easy to follow the article. The presented approach could be interested for the readers, however the results obtained cannot be considered as a breaking-through technology.
Responses1:
Thank you so much for taking the time to read the article and suggest revisions. The contributions of this paper include: we proposed a better event information visualization method; designed a target detection network model with balanced effects; produced two experimental data sets; verified that pseudo-frame images can efficiently complete target detection tasks. These have been covered in detail in the conclusion.
Based on your suggestion, it inspired me that I need to emphasize what is the contributions of this paper and the explanation and description of the experimental results in the paper are not clear enough, so I have made some modifications in the paper.
Some modifications are listed below:
1、I have highlighted the contribution of this paper in the Abstract. “Compared with methods with constant time interval and constant number of events, it can better convert event information into pseudo-frame images.” & “Compared with other algorithms, it has a more balanced detection effect.”
3、In section 5.3, I have added some descriptions about the comparison of the experimental results with other methods:
(1)Add “the AP of the image obtained by the ATR was 5.11% and 4.74% higher than that the method with CTI and CNE obtained, respectively.” to the first experimental analysis.
(2)Add “
- The model size is directly proportional to the mAP and inversely proportional to the FPS.
- The improved YOLOv1 and YOLOv2 have lower mAP than YOLOv3 and YOLOE,but have higher FPS.
- YOLOv3 has the highest mAP, the largest Model size, and the lowest FPS.
- YOLOE's mAP is 85.11%, 6.84% less than YOLOv3; YOLOE's FPS is 109, 32 more than YOLOv3; YOLOE's Model size is 163, 72 less than YOLOv3.
The improved YOLOv1 and YOLOv2 are suitable for applications that pay more attention to the detection speed and do not require high detection accuracy. YOLOv3 is suitable for applications that pay more attention to detection accuracy and do not require detection speed. YOLOE has a balanced performance in detection accuracy and detection speed, and is suitable for applications that have certain requirements for detection accuracy and detection speed. Therefore,” to the second experimental analysis.
Has been modified in the manuscript, please check.

Reviewer 3 Report
The authors propose an event information visualization method with adaptive temporal resolution capable of generating pseudo-frames for object detection. The also compare the proposed detection network (YOLOE) with adaptive temporal resolution to CTI and CNE.
The work is very interesting and as the authors mention, there is still only a few works in the area.
The authors mentioned that they had to create the datasets in order to validate the proposed network. Are the created datasets publicly available? This would be interesting if any reader intends to replicate or extend the current work.
What is the expected result of the proposed algorithm when detecting multiple objects in the same image? Consider images with objects superimposing each other (intersection/occlusion). I believe that the performance and accuracy of the technique will diminish a lot, since the pseudo-image is mainly based on the contours of the objects. What are the author considerations about that? This limitation of the proposed network should be described in the text.
According to Table 3, YOLOv3 showed the best mAP of all network models tested. This should be mentioned in the text. Also, the advantages of using YOLOE instead of YOLOv3 should be made clear in the text. When one technique is recommended rather than then other?
Some general comments and writing errors found are listed as follows.
"obtained respectively." -> "obtained, respectively."
"Dynamic Vision Sensor" -> "A Dynamic Vision Sensor"
"Real-time detection of objects becomes particularly" -> "real-time detection of objects particularly"
"can describe as:" -> "can be described as:"
"the less events captured by DVS The more, but the signal-to-noise ratio will be seriously reduced." -> please rewrite
"events generated" -> "events are generated"
which are the APS camera flaws?
"Then two decomposed" -> "Then, two decomposed"
"imagewhich" -> "image which"
please better explain how both brightening and darkening events may happen at the same position at same time. you may improve this text: "Considering that moving targets will generate brightening events at the front end and darkening events at the rear end of the sensor respectively, assuming that there is no influence of noise, the meaning of a yellow pixel is: both the front end and rear end of the captured object pass through the pixel in a time."
"(d) and e" -> "(d) and (e)"
"a lot defects" -> "a lot of defects"
"area. As shown" -> "area, as shown"
"Nearest neighbor filtering is to count the number N of real events at the adjacent pixel positions of the target pixel, and determine the value of the target pixel position by judging the size of N." -> this sentence is duplicated in the text
please be more clear when you say "such as the first and fourth images in 2)" and "such as the first and fourth images in 3)"
"On the one hand," -> "On one hand,"
"If the object moves" -> "if the object moves"
"Chapter summary" -> "Section summary"
"This chapter proposes" -> "This section proposes"
"Figure 10 the flowchart" -> "Figure 10 represents the flowchart"
"Generated iamge" -> "Generated image"
"Compared with the traditional target detection algorithm" -> what is the traditional algorithm?
"the better developed ones." -> "the best developed ones."
"CPU of i9-11900H and a GPU of RTX3070." -> "i9-11900H CPU and a RTX3070 GPU."
"in Chapter 3" -> "in section 3"
"data set 2," -> "dataset 2,"
"intuitive; Thirdly," -> "intuitive. Thirdly,"
"images; Finally," -> "images. Finally,"
"obtained respectively." -> "obtained, respectively."
"it may not even be applicable in some scenarios." -> please mention some example scenarios in which the proposed technique would fail
Author Response
Point1: The authors mentioned that they had to create the datasets in order to validate the proposed network. Are the created datasets publicly available? This would be interesting if any reader intends to replicate or extend the current work.
Response1: I can make publicly available datasets and I will submit them to journal editors.
Point2: What is the expected result of the proposed algorithm when detecting multiple objects in the same image? Consider images with objects superimposing each other (intersection/occlusion). I believe that the performance and accuracy of the technique will diminish a lot, since the pseudo-image is mainly based on the contours of the objects. What are the author considerations about that? This limitation of the proposed network should be described in the text.
Responses2:
Thank you so much for taking the time to read the article and suggest revisions. The expected result when there are multiple objects in an image is that each object can be detected accurately. When multiple objects are superimposed interactively, some features are hidden, and the detection effect will indeed be greatly reduced. I encountered the same situation during the experiment, just very sorry that it didn't occur to me that this situation should be explained in the manuscript. Designing a network with a stronger learning ability and preparing a larger and more complete dataset may reduce the impact of this problem, but that will slow down the detection speed. The research background of this manuscript is fast space target detection, and the phenomenon of overlapping targets is rare, so I balanced the detection speed and detection accuracy when designing the network.
Based on your suggestion, I have explained the phenomenon of multi-target overlap to some extent in section 6. “In addition, when detection backgrounds and detection targets change, the detection performance of the target detection model introduced in this paper will fluctuate greatly, because it is designed based on the research background of this paper. It may not even be applicable in some scenarios, such as an application scenario with huge and complex backgrounds, an application scenario focusing on fine textures of the target surface, and an application scenario with small and jumbling targets.”
Has been modified in the manuscript, please check.
Point3: According to Table 3, YOLOv3 showed the best mAP of all network models tested. This should be mentioned in the text. Also, the advantages of using YOLOE instead of YOLOv3 should be made clear in the text. When one technique is recommended rather than then other?
Responses3:
Based on your suggestion, I realized that there was not a complete analysis of the experimental results, so I have added the following to the manuscript:
(1)Add “the AP of the image obtained by the ATR was 5.11% and 4.74% higher than that the method with CTI and CNE obtained, respectively.” to the first experimental analysis.
(2)Add “
- The model size is directly proportional to the mAP and inversely proportional to the FPS.
- The improved YOLOv1 and YOLOv2 have lower mAP than YOLOv3 and YOLOE,but have higher FPS.
- YOLOv3 has the highest mAP, the largest Model size, and the lowest FPS.
- YOLOE's mAP is 85.11%, 6.84% less than YOLOv3; YOLOE's FPS is 109, 32 more than YOLOv3; YOLOE's Model size is 163, 72 less than YOLOv3.
The improved YOLOv1 and YOLOv2 are suitable for applications that pay more attention to the detection speed and do not require high detection accuracy. YOLOv3 is suitable for applications that pay more attention to detection accuracy and do not require detection speed. YOLOE has a balanced performance in detection accuracy and detection speed, and is suitable for applications that have certain requirements for detection accuracy and detection speed. Therefore,” to the second experimental analysis.
Has been modified in the manuscript, please check.
Point4: Some general comments and writing errors found are listed as follows.
- "obtained respectively." -> "obtained, respectively."
Response: Has been modified in the manuscript, please check.
- "Dynamic Vision Sensor" -> "A Dynamic Vision Sensor"
Response: Has been modified in the manuscript, please check.
- "Real-time detection of objects becomes particularly" -> "real-time detection of objects particularly"
Response: Has been modified in the manuscript, please check.
- "can describe as:" -> "can be described as:"
Response: Has been modified in the manuscript, please check.
- "the less events captured by DVS The more, but the signal-to-noise ratio will be seriously reduced." -> please rewrite
Response: Has been modified in the manuscript, please check.
- "events generated" -> "events are generated"
Response: Has been modified in the manuscript, please check.
- which are the APS camera flaws?
Response: Since the flaws of the APS camera are not mentioned in the article, I have changed “But it suffers from APS camera flaws.” to “But it suffers from APS camera flaws, such as redundant information, low time resolution, and low dynamic range.”
- "Then two decomposed" -> "Then, two decomposed"
Response: Has been modified in the manuscript, please check.
- "imagewhich" -> "image which"
Response: Has been modified in the manuscript, please check.
- please better explain how both brightening and darkening events may happen at the same position at same time. you may improve this text: "Considering that moving targets will generate brightening events at the front end and darkening events at the rear end of the sensor respectively, assuming that there is no influence of noise, the meaning of a yellow pixel is: both the front end and rear end of the captured object pass through the pixel in a time."
Response: I have changed the text to “Considering that the front end of the moving target will generate a brightening event, and its rear end will generate a darkening event, assuming there is no noise, the meaning of the yellow area is that both the front end and the rear end of the moving target pass through the same pixel position in a time.”. Has been modified in the manuscript, please check.
- "(d) and e" -> "(d) and (e)"
Response: Has been modified in the manuscript, please check.
- "a lot defects" -> "a lot of defects"
Response: Has been modified in the manuscript, please check.
- "area. As shown" -> "area, as shown"
Response: Has been modified in the manuscript, please check.
- "Nearest neighbor filtering is to count the number N of real events at the adjacent pixel positions of the target pixel, and determine the value of the target pixel position by judging the size of N." -> this sentence is duplicated in the text
Response: Has been modified in the manuscript, please check.
- please be more clear when you say "such as the first and fourth images in 2)" and "such as the first and fourth images in 3)"
Response: I have changed “the first and fourth images in 2)” to “the first images and the fourth images in (b)”, and changed “the first and fourth images in 3)” to “the first images and the fourth images in (c)”. Has been modified in the manuscript, please check.
- "On the one hand," -> "On one hand,"
Response: Has been modified in the manuscript, please check.
- "If the object moves" -> "if the object moves"
Response: Has been modified in the manuscript, please check.
- "Chapter summary" -> "Section summary"
Response: Has been modified in the manuscript, please check.
- "This chapter proposes" -> "This section proposes"
Response: Has been modified in the manuscript, please check.
- "Figure 10 the flowchart" -> "Figure 10 represents the flowchart"
Response: Has been modified in the manuscript, please check.
- "Generated iamge" -> "Generated image"
Response: It’s a mistake in figure 10, I have changed. Has been modified in the manuscript, please check.
- "Compared with the traditional target detection algorithm" -> what is the traditional algorithm?
Response: I have added an attribute “which sequentially uses algorithms such as HOG, LBP, and Harris to extract features from images, and classifiers such as SVM and Decision Tree to learn and classify features” to “the traditional target detection algorithm”. Has been modified in the manuscript, please check.
- "the better developed ones." -> "the best developed ones."
Response: Has been modified in the manuscript, please check.
- "CPU of i9-11900H and a GPU of RTX3070." -> "i9-11900H CPU and a RTX3070 GPU."
Response: Has been modified in the manuscript, please check.
- "in Chapter 3" -> "in section 3"
Response: Has been modified in the manuscript, please check.
- "data set 2," -> "dataset 2,"
Response: Has been modified in the manuscript, please check.
- "intuitive; Thirdly," -> "intuitive. Thirdly,"
Response: Has been modified in the manuscript, please check.
- "images; Finally," -> "images. Finally,"
Response: Has been modified in the manuscript, please check.
- "obtained respectively." -> "obtained, respectively."
Response: Has been modified in the manuscript, please check.
- "it may not even be applicable in some scenarios." -> please mention some example scenarios in which the proposed technique would fail
Response: I have added “such as an application scenario with huge and complex backgrounds, an application scenario focusing on fine textures of the target surface, and an application scenario with small and jumbling targets” behind the “scenarios”. Has been modified in the manuscript, please check.
